# The Transformation by Catalysis of Prebiotic Chemical Systems to Useful Biochemicals: A Perspective Based on IR Spectroscopy of the Primary Chemicals: I. The Synthesis of Peptides by the Condensation of Amino Acids

**Ragnar Larsson [1],\* and Abdul Malek [2]** 

[1]   Dept of Chemical Engineering, University of Lund Box, 124, SE 221 00 Lund, Sweden
[2]   Technologie DMI 980 Rue Robert Brossard, Quebec, QC J4X 1C9, Canada; abdulmalek@qc.aibn.com
\*    Correspondence: Ragnar.Larsson@chemeng.lth.se

**Abstract:** It is now widely speculated that life originated at the "Black Smokers" of the undersea hydrothermal vents, where conditions exist for the formation of the primary ingredients and their subsequent transformation to higher biotic species such as amino acids, alcohols, etc. Any possible routes for the prebiotic oligomerization of simple compounds like amino acids, necessary for cell formation, has so far not been well understood. However, Leman et al. recently reported that under standard laboratory conditions carbonyl sulfide (COS) can "mediate" the oligomerization of simple amino acids in moderate yield. COS being a well-known volcanic gas points to its possible role in prebiotic peptide formation in the environment of the hydrothermal vents. Based on a previously developed and tested model for selective (vibrational) energy transfer (SET), we show that a COS-catalyzed condensation of $\alpha$-amino-acids can lead to the formation of polypeptides. We also indicate that other agents can act as catalysts of the amino acid condensation, such as $Fe(CN)_6^{3-}$ and cyanamide ($H_2N$-CN). This is related to the existence of vibrations with a frequency near to that of the critical vibration of the reactant, $\varrho_w$ ($NH_2$). This wagging vibration occurs at $1048 \pm 10$ cm$^{-1}$ (the mean value of Cu and Ni complexes) and, as the vibration of the presumed catalyst lies at 2079 cm$^{-1}$, one notes that one quantum of the catalyst equals two quanta of the $NH_2$ wagging: $2079/2 \times 1048 = 0.9919$. This is a good indication of a resonance.

**Keywords:** catalysis; amino acids; polypeptides; vibrational resonance; selective energy transfer (SET); prebiotic reactions; reorganization of orbital patterns (sp3 to sp2)

## 1. Introduction

Among the chemical reactions of direct importance for life is the condensation of amino acids to give polypeptides or (depending on size) proteins. By "condensation" is meant the type of reaction where two molecules are joined by the expulsion of a water molecule and the formation of a peptide bond: R1–NH–C=O–R2.

Such "condensations" are considered as part of the first steps in creating life from organic material [1–3]. After repetition of condensations between large numbers of amino acids, polymers such as proteins are formed.

These processes are usually quite slow because of the unfavorable thermodynamic and kinetic parameters for the reactions [4,5]. Therefore, a catalyst (or an enzyme) is needed to speed up the reaction. In this paper we will concentrate our attention on one of the first catalysts that acted under prebiotic conditions—namely, carbonyl sulfide (COS).

This is not an arbitrary choice; rather, we chose COS for two reasons, the first being the very strong infrared absorption of one of the three fundamental vibrations, at 2079 cm$^{-1}$ [6,7], and the consequential strong emission [8] of importance for the catalytic process; the other reason is the appearance of COS in volcanic eruptions [9], which in turn gave the catalyst a load of energy from the start.

Such a reaction must have been conditioned by taking place in confined and restricted environments (e.g., the hydrothermal vents in mid-ocean ridges discovered in the 1970s) where selective catalyzed chemical reactions could proceed, forming progressively higher oligomeric chemicals. This notion is supported by the recent discovery that the earliest life forms on Earth known so far are putative fossilized micro-organisms [10] found in hydrothermal vent precipitates in Québec, Canada, dated from 3.77 to 4.28 billion years old.

As is now known, life is functioning through the specialized chemistry of a limited number of elements, such as hydrogen, carbon, nitrogen, oxygen, sulfur, phosphorous, and a few transition metals and their compounds. Aside from water, the physicochemical processes of life are largely based upon five key families of chemicals: lipids (fatty cell walls), carbohydrates such as sugars and cellulose (energy source structure elements), amino acids (protein metabolism), nucleic acids (self-replicating DNA and RNA), and some minerals.

In the 1950s it was discovered how one of these "families", viz. the amino acids, could result from purely inorganic substances such as water, nitrogen, ammonia, and carbon dioxide. In a classical experiment by Stanley Miller [11,12] it was shown that amino acids could be formed from such an inorganic mixture if it was exposed to an electric discharge. This was an important step towards the understanding of how life came about, but provided no clue as to how oligomers like polypeptides or proteins could be formed.

No other viable processes for the formation of prebiotic oligomers under ambient natural conditions are known. Therefore, we now set out to try a new concept of catalysis, developed by one of us [13]—"selective energy transfer" (SET)—using the catalyst COS, described above.

## 2. The Selective Energy Transfer (SET) Theory

During the past few decades, one of the present authors has developed a new model for catalysis, different from those schemes that have commonly been used to describe the technological aspects of a catalyst and the mechanism of a catalytic reaction.

The basis of this concept [14,15] is the selective transfer of energy from a molecular vibrator in the catalyst system towards a vibrating system in the molecule that is involved in the reaction. Because of this view, the new model has been named "SET" or the selective energy transfer model.

The main idea of the SET theory is that energy is transferred from a molecular vibration of the catalyst system to a vibration of similar frequency in the reacting molecule. The condition for such a transfer of energy is that a state of resonance holds between the two systems. The rate of energy transfer from one vibrator to another in a damped, coupled oscillating system was calculated in classical physics [16].

The classical expression for the rate of energy exchange is

$$P = P_{res} \times (\omega^2/\tau^2)/[(v^2 - \omega^2)^2 + \omega^2/\tau^2], \tag{1}$$

where $P$ is the energy transfer per unit of time;

$P_{res}$ is the power absorption at resonance (i.e., when $v = \omega$);

$\omega$ is the frequency of the catalyst system;

$v$ is the frequency of the reacting molecule;

$\tau$ is the relaxation time; and

$Q = \tau \times v$ is the "quality factor".



We take this rate—after integrating over all possible values of the "quality factor" Q, and freshening it up to quantum chemistry standards—to equal the rate of the chemical reaction [13,15]. Further details can be found in [13].

The SET theory has recently been summarized [15]. One fact, illustrated by the investigated catalytic systems, is that the activation energy (enthalpy) is built up by a certain number of vibrational quanta related to the $\upsilon$ vibration. This can be expressed [15] as a second-order relation between the activation enthalpy, $\Delta H^{\#}$, and the vibrational quantum number n:

$$\Delta H^{\#} = M0 + M1\,n + M2\,n^2. \tag{2}$$

The second-order term corresponds to the anharmonicity of the system, and shall consequently have a negative sign. The M1 coefficient describes the wave number of the $\upsilon$ vibration. M0 indicates the state of adsorption of the reacting molecule. In most cases reported before [15], M0 is very close to zero, indicating that the activation occurs for only lightly adsorbed species, but close to the catalyst surface (for reaction with another reactant available at the surface). This latter statement is a reasonable one, as the molecule can better keep its state of activation when not in contact with a solid surface. In the concluding remarks on this model [15], it was reported on the decomposition of formic acid (which can react along two different lines), but also on some dehydrochlorination reactions. Recently, the SET model has been applied to the technically important dehydrogenation of propane [17], as well as to the equally important ammonia synthesis from $N_2$ and $H_2$ [18]. In all these cases, one can state that the SET model works well [19].

*2.1. Peptide Bond Formation:*

As mentioned in the Introduction, we concentrate here on the formation of the polypeptides and proteins, because of their fundamental importance in life processes. Although the formation of amino acids under reductive atmosphere has been achieved [11], the second sequence (i.e., the formation of polypeptides [3]) seems to be a much more formidable task. The formation of the peptide bond under thermodynamic control is not favorable [4,5], compared to the free energy of hydrolysis of an internal peptide bond with $\Delta G \sim -2$ to $-6$ kJ mol$^{-1}$.The formation of long peptides at realistic concentrations of $\alpha$-amino acids remains highly unfavorable [4,5]. It is therefore essential that this overall free energy cost must be compensated by coupling the formation of each peptide bond by an activating agent to make the reaction favorable. In addition, a metal ion to "mediate" this process into an integrated whole would seem ideal for the formation of polypeptides under the hydrothermal condition of the undersea vents.

In the present proposal, the $-NH_2$ group of one amino acid moiety and the $-COOH$ groups of another one are held in close proximity by, for example, a Ni (II) or a Cu(II) ion (as shown in Scheme 1). As discussed further on in Section 2.2, the COS molecule transfers quanta of vibrational energy to the coordinated $-NH_2$ group through a matching resonance vibration, leading to the condensation reaction involving the nearby $-OH$ group, thus forming the peptide link, $-(C=O) - NH-$, and releasing a molecule of water. The COS molecule delivers the energy necessary for this reaction, "without itself being consumed thereby" [20,21].

One of the acids is coordinated in a true fashion to the metal atom ( M ) as a chelate, whereas the other one binds strongly only via the NH₂ group. This implies that the C=O (OH) group is intact so that it can react with the H atom , set free by the NH₂ wagging. As described in the text, the catalyst activates the out of plane wagging vibration of the NH₂ group ( of the "R1-compound" ) so that one hydrogen atom is let free to attack the OH group of the "R2 compound". These groups, are shown in the figure by a triangle with red, broken lines.

The resulting peptide group is indicated by red colors in the lower drawing.

**Scheme 1.** "Condensation" of two amino acids catalyzed by COS.

In view of these results, one can ask, how did the SET model work when meeting the demands of life-creating reactions some billions of years past (and possibly continuing even today)?: It is now generally assumed [10] that the first and most primitive cells (prokaryotes) were formed in the niche environments of continuous supply of primary chemical compounds. However, what can we tell about the role of the amino acids when building the biologically important structures of multi-peptides or possibly proteins?

In the present investigation, we used SET to try to answer these questions and speculate on the possible formation of simple proteins based on reactions of amino acids under catalytic conditions. We found it necessary here to include, besides COS, a water-soluble salt of a $2^+$ - metal ion. However, the burning question is, how did this possibly catalytic reaction manifest itself? What happened? Was COS a true catalyst, or was it consumed in the reaction as assumed by Leman et al. [22]? This is good ground to test the SET theory for catalysis.

## 2.2. COS and Resonance Conditions

As mentioned in the Introduction, the basis of SET [13] is that energy is transferred between catalyst and reactants when there is a state of resonance between some vibrations of these species. In turn, this requires that there is a simple relation between the vibrational frequencies of catalyst and of the reactant-to-be. Thus, $\omega/\upsilon$ can be 1:1, 2:1, 1:2, 3:1, etc. The important point is that an equal amount of energy is donated from the catalyst as is accepted by the reactant molecule.

In Table 1, we have collected vibrational frequencies of the molecule COS [7] as well as of a small amino acid (glycine) bound to a metal atom (in this case $Ni^{2+}$ or $Cu^{2+}$) [23]. At first glance one does not note any obvious relation between the wave numbers of COS and those of the metal complexes of the amino acid. However, one clearly notices, vide Table 2, that two times of the 1058 $cm^{-1}$ vibration of the (Cu) amino acid complex agrees well with the $\upsilon$ 3 vibration of COS (2079 $cm^{-1}$).

**Table 1.** Vibrational assignments for Ni and Cu glycinate complexes [23] and for carbonyl sulfide [7]. Assignments are denoted as in these references. Vibration frequencies are given in $cm^{-1}$. Data used by copyright according to Sections 107 and 108 of the 1976 United States Copyright Act (relating to research use/"fair use").

| Assignment | Cu (Glycinate)$_2$ | Ni (Glycinate)$_2$ | Assignment | COS |
|---|---|---|---|---|
| $\upsilon$ (NH$_2$) | 3320 | 3340 | | |
| $\upsilon$ (NH$_2$) | 3260 | 3280 | $\upsilon$ 1 | 850 m |
| $\upsilon$ (C=O) | 1593 | 1589 | | |
| $\delta$ (NH$_2$) | 1608 | 1610 | $\upsilon$ 2 | 527 m |
| $\upsilon$ (C–O) | 1392 | 1411 | | |
| $\varrho_t$ (NH$_2$) | 1151 | 1095 | $\upsilon$ 3 | 2079 v s |
| $\varrho_w$ (NH$_2$) | 1058 | 1038 | | |
| $\varrho_r$ (NH$_2$) | 644 | 630 | | |
| $\delta$ (C=O) | 736 | 737 | | |
| $\prod$ (C=O) | 592 | 596 | | |
| $\upsilon$ (MN) | 439 | 439 | | |
| $\upsilon$ (MO) | 360 | 290 | | |

**Table 2.** Collection of a series vibrational frequencies ($cm^{-1}$) from both an amino acid chelate (designated "Cu") and the COS molecule. The percentage of the differences are calculated based on the data in column 2.

| Number | Cu (Glycinate)$_2$ | Ways of Forming Differences | COS | Diff (Cu - COS) $cm^{-1}$ | Diff Absolute Numbers % |
|---|---|---|---|---|---|
| 1 | 1593 | $\upsilon$ C - $\upsilon$1(COS) | 850 | 743 | 46.6 |
| 2 | | $\upsilon$Cu - 2 × $\upsilon$1(COS) | | −107 | 6.7 |
| 3 | 1608 | $\upsilon$Cu - $\upsilon$1(COS) | 850 | 758 | 47.1 |
| 4 | | $\upsilon$Cu - 2 × $\upsilon$1(COS) | | −92 | 5.7 |
| 5 | 1392 | $\upsilon$Cu - $\upsilon$1(COS) | 850 | 542 | 38.9 |
| 6 | | $\upsilon$Cu - 2 × $\upsilon$1(COS) | | −308 | 22.1 |
| 7 | | $\upsilon$Cu - 2 × $\upsilon$2(COS) | 527 | −338 | 24.3 |
| 8 | | $\upsilon$Cu - 3 × $\upsilon$2 (COS) | 1581 | −189 | 13.6 |
| 9 | 1058 | $\upsilon$Cu - 2 × $\upsilon$2 (COS) | 2 × 527 | 4 | 0.4 |
| 10 | 1058 | 2 × $\upsilon$Cu - $\upsilon$ 3(COS) | 2079 | 37 | 3.5 |

This observation gives 2 × 1058/2079 = 1.018, which is as close to a resonance condition as one might imagine. However, this amino acid vibration at 1058 $cm^{-1}$ is not just any vibration. It is the out-of-plane wagging of the amino group (cf. Figure 1), and designated as $\varrho_w$ (NH$_2$), [23], Table III -19 (page 236). For the nickel complexes, the similar value for the NH$_2$ wagging is 1038 $cm^{-1}$ [23], which gives 2 × 1038/2079 = 0.999. Thus, taking the mean value of both Cu and Ni data, we get *$\omega$ (COS)/2 $\upsilon$ glycinate = 1.009 ± 0.01.*

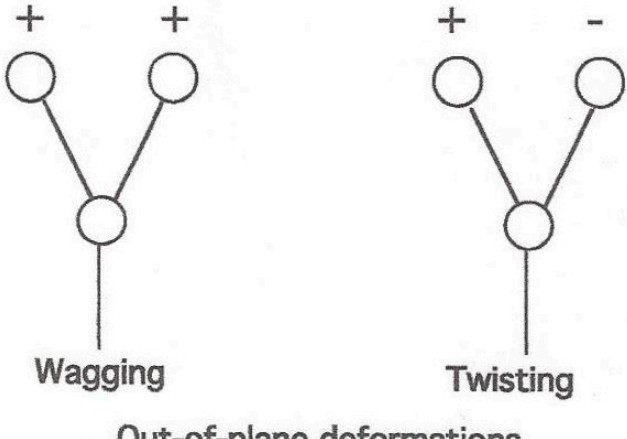

**Figure 1.** Schematic representation of the out-of-plane vibrations in relation to the plane formed by three atoms.

We can thus write: $\omega = 2\,v$, or

$$\omega/v = 2{:}1. \tag{3}$$

Relation (3) implies that two vibrational quanta of the $\varrho_w$ ($NH_2$) vibration of the amino acid can accept energy from one quantum of the COS molecule. As Table 2 indicates, another combination of vibrations is close to a state of resonance, viz., $2 \times v_2$ of COS agrees quite well with 1058 cm$^{-1}$ of the Cu chelate (i.e., $2 \times 527/1058 = 0.996$). However, we prefer to concentrate the discussion to the above treatment of $v\,3$ of COS because this vibration shows a very strong absorption of IR radiation [6], (vide Figure 1), and thus gives a much higher emission than what the $v\,2$ of the same molecule. In other words, the flow of quanta from 2079 cm$^{-1}$ is much stronger than the flow of quanta from 527 cm$^{-1}$.

The designation of this vibration as $\varrho_w(NH_2)$ means – as said above - that it is an out-of-plane "wagging" movement, in which both H atoms bend away from the plane that they form with nitrogen (cf. Figure 2) [24].

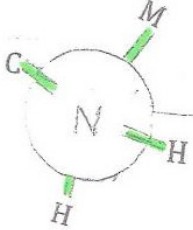

**Figure 2.** Environment of the nitrogen atom with sp3 orbital hybridization (cf. Scheme 1). M represents the metal ion that keeps the two amino acid molecules together, C depicts the carbon atom nearest the carboxylate group of the acid under consideration.

It follows from SET that the vibration found in this supposed molecular reaction is the one that actually activates the reaction [13–15]. If activation energies had been measured, we would have seen (Equation (1)) that they all had been a sum of the energy of a vibration with a frequency of 1058–1038 cm$^{-1}$. Of course, this is an approximate statement, as one must take the anharmonicity into consideration (Equation (2)).

In the present case, the $NH_2$ group is part of an amino acid chelate around the metal ion M. To emphasize the role of the $NH_2$ group, we display a schematic picture of the nearest atom groups around the tetrahedrally bonded central nitrogen atom (Figure 2).

The tetrahedral configuration around the nitrogen atom can be described so that each of the four groups at the corners of the tetrahedron engages with one sp3 orbital, each one originating from

the nitrogen atom. However, when the two hydrogen atoms move together, in the $\varrho_w(NH_2)$ vibration, perpendicularly against the N-H-H plane, the type of bonding tends more and more towards an sp2 configuration. In the extreme, the system forms three sp2 orbitals from the central nitrogen atom towards each of three remaining atoms or atomic groups (cf. Figure 3). This implies that one of the two hydrogen atoms must leave the scene. Or rather (vide Scheme 1), this H atom is free to attack the OH- group of the neighboring amino acid, forming water and making possible a combination of the two amino acids to form a peptide bond.

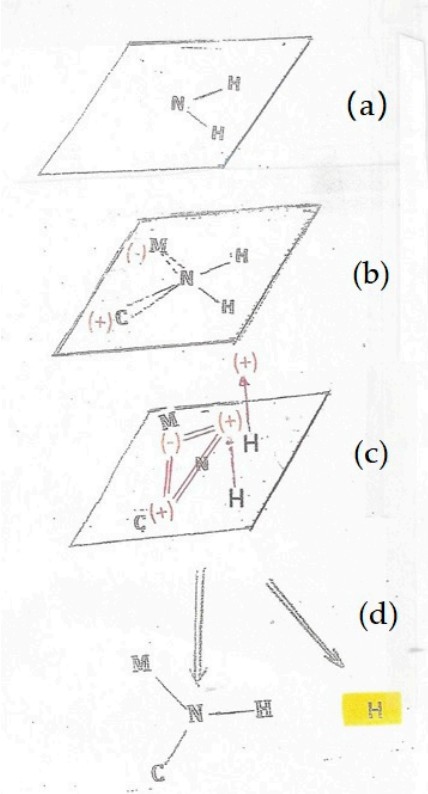

**Figure 3.** (**a**) **The plane of wagging.** The three atoms, N, H, and H, define a plane in space. When the two hydrogen atoms move perpendicularly against this plane, the vibration is called "wagging", cf. Figure 2. (**b**) **The more complete structure around the sp3 nitrogen.** The tetrahedral structure of the nitrogen (indicated in Schemes 1 and 2) is made up by one meatal atom (M) and one carbon atom (C), next neighbor to the carboxylate group (cf. Figure 4). In order to indicate the spatial (tetrahedral) arrangement, a plus sign indicates that the atom group is positioned above the NHH plane. Likewise, a minus sign indicates a position below that plane. (**c**) **The wagging effect.** The figure illustrates how the two H atoms rise above the NHH plane (also here indicated by plus-sign and in red color) at a highly activated wagging vibration. One can also easily distinguish that the left-side hydrogen atom can form part of a tilting CHM plane into which the nitrogen atom can intrude. (This plane is indicated by double-drawn red lines between the atoms concerned.) The other H atom is distinctly placed outside the tilting CHM plane. As described in the text, this means that the N atom develops three sp2 orbitals that keeps the CHM plane together. (**d**) **At high excitation, the nitrogen atom will be unable to bind four atoms.** The consequence of this will be that one hydrogen atom (marked in yellow) is set free.

Thus, the interference of the catalyst (COS) and the amino acids results in a peptide bond between the two molecules and the formulation of a water molecule. The catalyst COS remains intact, but for a loss of some energy and ready for the next cycle of activation and energy transfer under the hydrothermal vent condition (along with fresh supply of COS coming from the effluent of the vents); before being lost due to dissipation or through hydrolysis or other types of reactions.

Scheme 1 illustrates that the role of the metal ion—besides achieving the best possible resonance conditions—is to create order between the two amino acid molecules; we name them as amino acid No. 1 and amino acid No. 2, respectively. The importance of metal atoms being present has been emphasized experimentally by Huber, Wächtershäuser, Eisenreich, and Hecht [25,26]. One might note that the incoming amino acid (No. 2) has supposedly not formed a regular chelate, but hinges on the strong M–N bond, thus exposing its OH group for attack from the H atom of the "No. 1" $NH_2$ group. As it is supposed that this reaction was happening at the verge of a volcanic vent, where metal sulfides (FeS or NiS [26]) had been precipitated over time, it is no surprise that the possibility of bonding ligands to a free metal ion was severely sterically limited. Therefore, the unbonded C–OH group swings open to an attack from a free H atom (Figure 4).

When all this has happened, two alternative reactions might further take place: either the metal–carboxylate bond can split and a dimer, mono-peptide will form (Scheme 1), or the system keeps the original metal–carboxylate bond intact and forms a new chelate—this time with the recently formed M–$NH_2$ group as the bonding partner (vide Scheme 2). The approach of a third amino acid makes possible the creation of a second peptide group. In principle, this procedure can go on in many steps, thus forming polypeptides. Such molecules are considered to be important building blocks for the emergence of life (e.g., [5]).

One crucial question remains to be commented upon: Why is it only one of the two amino groups in Scheme 1 that responds to the infrared radiation that emits from the COS catalyst? Obviously, it is the one –$NH_2$ group that first responds to this radiation that we observe, but why is there a difference? We suggest that the amino acid that reacts is the one that has a –COOH group firmly bonded to the metal atom as $COO^-$. It is the –NH2 group of the strong chelate that responds to the IR radiation and consequently releases one hydrogen atom. It is this free hydrogen atom that attacks the –OH group of the other amino acid. This is lightly indicated in the drawings of Scheme 1, and made more visible in Figure 4. Thus, it seems that a metal atom is crucial in this context [25], giving a strong resonance between the vibrational frequencies.

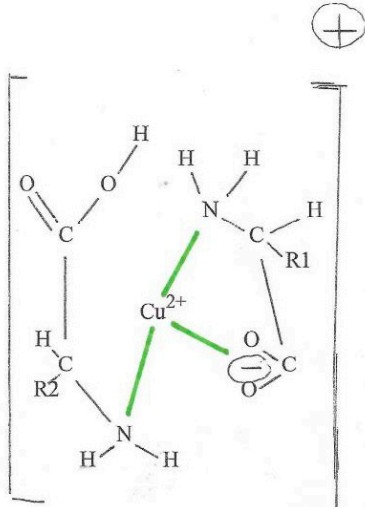

**Figure 4.** The situation of binding between one copper ion and two amino acids just before the reaction to form a peptide bond. The strong bonds are designated in green color.

**Scheme 2.** Continued "condensation" of another amino acid ($R_3$) with the product, the formation of which was described in Scheme 1. The resulting peptide is marked in red and one notes that the substituents are found in the $R_1$, $R_2$, $R_3$ order. Furthermore, the carboxylate end of the chain belongs to the first amino acid ($R_1$) and the amino end belongs to the last one ($R_3$).

## 3. Further Tests of SET Catalysis

The presentation above, centering on the activating effect of COS, has shown that the important piece of the reaction is the vibrational excitation of the $\varrho_w$ ($NH_2$) vibrational mode of an amino acid, bonded to a metal ion. To find other activating agents with the same ability, one must then look for substances with some group of atoms that can vibrate with frequencies approaching resonance with the $\varrho_w$ ($NH_2$) vibration. In the following, we discuss two such systems.

### 3.1. Iron(II/III) Hexacyano Complexes

The investigation of Leman et al. [22] gives further information that might guide us in our search towards other examples of SET. One first notes that for most of the investigated systems, the time to achieve results is measured in hours if not in tens of hours. (Cf. Table 1 of Ref. [22]). The only drastic exceptions are those experiments where hexacyanoferrate(III) is used as a reactant. For these systems, conversions of about 75% were obtained in 5 min, 20 min, and 1.5 h.

The quoted authors suggest [22] that the iron (III) oxidizes the $\alpha$-amino acid thiocarbamate, considered to be an intermediate in the formation of dipeptides. Of course, in this process, the iron (III) is reduced to iron (II).

Regardless, we find from Table III.34 of Ref. [23] that the strongest of the C–N stretching modes of the Fe(III)complex ($F_{1u}$) appears at 2118 cm$^{-1}$, whereas the same for the Fe(II)complex is 2044 cm$^{-1}$. Division of these values by two yields 1059 cm$^{-1}$ for Fe(III) and 1022 cm$^{-1}$ for Fe(II). These values are critically close to those of the amino acid complexes [23] discussed above, viz. 1058 cm$^{-1}$ (for Cu) and 1038 cm$^{-1}$ (for Ni). "Close" in the sense that in this case the catalyst splits its vibrational quantum into two equally large bits, equal to 1059 cm$^{-1}$ for Fe(III) resp. 1022 cm$^{-1}$ for Fe(II).

Whether reduced or not, it seems from the results of Orgel et al. (Table 1 of Ref. [22]) that it is the hexacyanoferrate(II) or (III) that form the best catalysts [22]. If we apply the SET model, the time

that is needed to stimulate the $\varrho_w$ (NH$_2$) vibration of the amino acid is important. As stated above, the previous authors [22] argued that the role of Fe(CN)$_6{}^{3-}$ was to oxidize sulfur-containing species and in this way come forward to the amino acid N-carboxy anhydride (NAC) that was considered as the route towards peptide formation. However, in regard to the very close to equal agreement for the vibrational data given above (i.e., 1058 (amino acid complex) and 1059 cm$^{-1}$ (hexacyanoferrate), respectively), we are tempted to state that the active catalyst is hexacyanoferrate(III) interacting with a copper(II) chelate that in one or possibly two steps makes the reaction possible.

### 3.2. Cyanamide Test

Stanley Miller left behind not only his famous report [11,12] on the formation of amino acids by the exposure of simpler substances (e.g., CH$_4$, NH$_3$, etc.) to electric discharges. He also left many vials containing the results of other investigations. Most of these samples were not analyzed until the first decades of the 21st century. Then, some of his previous collaborators found the remains and started to publish the content of Miller's notebooks [27].

The strategy of Miller as well as his followers was to combine the well-established amino acid synthesis by electric discharge with a parallel exposure of the reaction mixture to cyanamide, H$_2$N–CN [27]. This substance was added intermittently to the reacting gas mixture [27]. The result was that many amino acids were formed (probably by the electric discharge effect), along with dipeptide combinations of many of these acids (e.g., gly–gly, gly–ala, glu–gly).

We will now test if the SET model works for these reactions equally well as in the previously discussed cases. For this purpose, we use glycine as the model substance. Our starting view is that the reaction proceeds in two stages: the first reaction is the formation of amino acids as found by Miller [11], the second reaction is the formation of dipeptides—very much like what is described above in Section 2. In order to follow this plan, we first need to know infrared data for cyanamide [28], and secondly infrared data for (free) glycine [29]. Thus equipped, we suggest following the path of the procedure applied in the above chapters. This means that we give the major role to the $\varrho_w$(NH$_2$) vibration of the glycine molecule, and start looking for a counterpart in the spectrum of the cyanamide molecule. Such comparisons are made in Table 3.

**Table 3.** Comparison between vibration frequencies of $\varrho_w$(NH$_2$) of glycine [29] and those cyanamide vibrations [28] that best agree with the glycine vibration. The difference between columns 5 and 4, in absolute numbers (Diff), is expressed as per cent of the "resulting product". Copyrights for these data relate to those Of Ref [28] and Ref [29].

| Conditions | Glycine Vibration cm$^{-1}$ [29] | Factor | Resulting Product cm$^{-1}$ | Cyanamide Vibration cm$^{-1}$ [28] | Diff% |
|---|---|---|---|---|---|
| Acid | Not present | | | | |
| Neutral | 882 (m) | 1 | 882 | 937 (v w) | 6.2 |
| | | 2 | 1764 | 1576 (m) | 10.7 |
| | | 3 | 2646 | 2731 (v w) | 3.2 |
| Alkaline | 1069 (w) | 1 | 1069 | 1094 (w) | 2.3 |
| | | 2 | 2138 | 2199 (v w) | 2.9 |
| | | 3 | 3207 | 3261 (v s) | 1.7 |

The winner in this contest is the 3261 cm$^{-1}$ vibration of cyanamide. As all other of the tabulated vibrations are denoted as weak or possibly medium, this very strong absorber—and consequently, very strong emitter—will do a good job of supplying the glycine vibrator with many quanta that will stimulate the NH$_2$ wagging and consequently work for the expulsion of one of the two hydrogen atoms, as described in Section 2.

Following the treatment of the previous systems, COS and ferricyanide complexes, we find that one quantum of the excited "activating agent", 3261 cm$^{-1}$, can be compared with three quanta of the NH$_2$ wagging of glycine (1069 cm$^{-1}$; cf. Table 3).

As $3 \times 1069 = 3207$, we get a measure of a possible resonance as $3261/3207 = 1.017$, which is as good as the corresponding ratios reported above. Thus, we can confirm the reasoning of Stanley Miller, given that the prebiotic atmosphere contained $H_2N\text{-}CN$.

One can further note that, as Parker et al. [27] pointed out, the reactive amino acid species is $H_3N^+C(RR')\,COO^-$. This means that, in the present case, a proton is substituting the metal ion in the reactions of Schemes 1 and 2 in our discussion above. Thus, the transformation sp3 $\geq$ sp2 is still the central one in the catalytic reactions. On the other hand, in all the cases where $H_2O$ is formed from the interaction of H and C–OH, it is necessary that the pH is well below the isoelectric value so that there are undisturbed OH-groups present that can participate in the reaction stoichiometrically.

## 4. Conclusions

In the above chapters we described routes towards the formation of biomolecules using the SET model for catalysis.

As stated in the Introduction, there are quite a few examples of the successful use of SET under present-day laboratory conditions. On the other hand, we do not know exactly what happened many billions of years ago. Hence, we now propose that the formation of long chains of polypeptide molecules can be explained by the application of SET to available catalysts and activating materials.

The gaseous substance of COS works very well and in accordance with the definitions by Berzelius and Ostwald: it is involved in the previously mentioned part of the reaction chain, but it is not consumed during the reaction [19,20]. One can note that—also in this case—the catalytic reaction mechanism works directly towards the desired product with limited amounts of intermediates. In contrast, the reaction scheme suggested by the previously mentioned authors [22] presupposes many intermediates, and consequently, a series of coupled reaction steps. These workers used large molar excess (8–16 times) of COS compared to amino acid concentration, which is an unrealistic circumstance near the hydrothermal vents. In our view, the use of large excess of the catalyst actually impeded the oligomerization process, giving poor yield and only giving low-chain oligomers because of the waste of the catalyst (COS) through stoichiometric reaction to form dead-end intermediates with the available amino acids or by forming $H_2S$ through hydrolysis that may have negated the desired oligomerization reaction.

The reaction series as depicted in Scheme 2 can thus be taken as a rapid route for molecular combinations that leads to stable states, suitable for processes that we recognize from present day's biochemistry.

**Author Contributions:** Conceptualization, R.L. and A.M.; Literature search and evaluation, A.M.; Calculations, R.L.; Methodology, R.L.; Writing—original draft, A.M. and R.L.; Data collection and evaluation, A.M. and R.L. All authors have read and agree to the published version of the manuscript.

**Funding:** This work, including the Open Access, was funded by the University of Lund.

**Acknowledgments:** The authors are very grateful for the support for this work provided by the University of Lund and the Department of Chemical Engineering of LU. This support was aimed at covering the costs of publishing the results of the study in full Open Access style. One of us (Ragnar Larsson) strongly thanks his daughter, Karin Lilja, for good guidance in computer use. Likewise, we, the authors, want to express our gratitude to Mr. Farhan Zaman of Ottawa, Canada, for his computer-related help.

**Conflicts of Interest:** The authors declare no conflicts of interest.

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
