# Peer review of "The Transformation by Catalysis of Prebiotic Chemical Systems to Useful Biochemicals: A Perspective Based on IR Spectroscopy of the Primary Chemicals: I. The Synthesis of Peptides by the Condensation of Amino Acids"

_applsci, doi:10.3390/app10030928_

Round 1
Reviewer 1 Report
The manuscript "The transformation by catalysis of prebiotic chemical systems to useful biochemicals: A perspective base on IR spectroscopy of the primary chemicals. I. The Synthesis of Peptides by Condensation of Amino Acids." By R. Larsson and A. Malek contains an interesting working hypothesis. Namely, that certain pre-biotic catalysts transferred their vibrational energy to amino acid complexes facilitating the amide bond formation in water and the subsequent appearance of peptides and proteins on Earth.
The starting point is good, since the authors found that aminoacids’ NH2 vibrations are coupled to those of the catalysts, and therefore the transfer of energy from the catalyst to the amino acid reacting moiety is possible. However, there is no more data or experiments to support this hypothesis. Key structural and mechanistic explanations suggested by the authors are carried out by drawings, without any molecular model, and do not provide any useful information. The idea that a H+ is released from M-NH2-R prior to the reaction with C(O)OH-M to form the peptide bond is interesting, but unsupported. No computational study is carried out. The hypothesis has not been minimally checked, and therefore it remains a starting point, without any contribution to the knowledge of this issue. Extensive experimentation or theoretical work is required prior to publication.
Author Response
Dear Referee 1,
Thank you for your thorough analysis of our manuscript.
There are, however, a few points that you raise, that must be further discussed.
You write that “the amino acids’ NH2 vibrations are coupled to those of the catalysts…” This does not necessarily hold for all NH2 vibrations. But what is interesting in this case is that one of them, especially when the amino acid is bonded to a metal ion shows out-of-plane wagging vibrations such that one vibration quantum of the catalyst interacts with two quanta of these wagging vibration. I have tried to emphasize this in the manuscript by using a somewhat larger style when writing the statement “it is not just any vibration” ( six lines above the presentation of equation (3) ).
It is the activation of this vibration and no other that makes the two hydrogen atoms in the NH2 group move in such a way that the orbital picture turns from sp3 toward sp2 . This, in turn, makes it impossible for the N-atom to bear more than one hydrogen atom. The other hydrogen is left as a neutral H atom that is caught by the OH group of the other amino acid. Because both molecules are closely coordinated to a metal ion ( illustrated in Figure 5 ) the transfer of a homeless hydrogen atom to a very closely situated OH group is quite a natural move. One does not have – as you do - to introduce free protons ( H+) here. This ordering effect by the presence of a metal ion was already pointed out by Huber et al, our ref [25-26] .
So far, I suppose we can agree ( but for the proton ).
Then you, mr/ms Referee present a lot of questions on the logic and philosophy of our methods. Let me try to answer them in order:
1.
You write: “However, there is no more data or experiments to support this hypothesis.” ( I might prefer to write ’there are no data’ , but that is a minor point.) Let me state that the conclusions we draw in this article, are based on OBSERVATIONS made before us, OBSERVATIONS on infrared spectra in general and on carbonyl sulfide especially made (or collected and commented upon) by the Nobel Laureate Gerhard Herzberg, OBSERVATIONS similarly done in the youth of coordination chemistry by professor K Nakamoto, specialist on IR spectra of coordinated ligands, on spectra of amino acids coordinated to metal ions ( Cu2+,Ni2+,etc ).
The OBSERVATIONS made by researchers as Huber and Wächtershäuser are already mentioned. Our conclusions are, of course, also related to the general law of physics.
It seems difficult to know what other experiments you think of.
1.a.
In this connection we can illustrate the discussion by mentioning the idea that the world is round : This thought is ascribed to the Greek philosopher ( scientist) Eratosthenes, 276 - 194 before our time. By OBSERVATIONS of the position of the sun in two cities at the same time he could calculate the periphery of the earth to 39360 km ( as compared to the present day value of 40 000 km ).
This concept, that the earth was round, was cherished by most educated people up to and past the middle ages. It was guiding Christopher Columbus and it served well. With this guiding principle, e. g. , a new continent was discovered.
The “final experiment” to decide was made by Fernando Magellan, whose ships left westwards and returned from the east ( only one ship).-----This example will show that ideas, founded on valid observations , are good ideas and can serve for centuries without a verifying experiment.
2.
You complain about the fact that we (the authors ) use drawings instead of molecular models to emphasize our findings. I fail to see the importance of the difference. (I will try to improve the clearness of Fig 4, which, I admit, is low).
3.
You write, in regard to the peptide formation ( already discussed in section 1 above ), that the transfer of a hydrogen atom is unsupported. Our point is that when the out-of-plane NH2 vibration is highly activated, the positions of the four atoms around the nitrogen atom are no longer well described by a tetrahedron. Rather the position of three of them tend to be best described as a plane (including the nitrogen atom).
As hinted at in the introductory chapter here, the orbital pattern around the N atom is then best described as sp2 and not as a sp3. The consequence of this is that one of the hydrogen atoms leave for assigning to the OH group of the other, nearly bonded, amino acid.
I hope we can agree that this atomic rearrangement is supported by present-day physical theories.
All in all - taking my above comments into consideration – I hesitate to make any changes in the manuscript. I sincerely hope that you will say “OK” to this.
Reviewer 2 Report
The manuscript by Ragnar Larsson et al., reports a possible routes for prebiotic oligomerization of simple compounds like amino acids, necessary for cell formation, considering carbonyl sulfide (COS) can ‘mediate’ the oligomerization of simple amino acids in moderate yield. In this framework, the possible role of COS in prebiotic peptide formation in the environment of the hydrothermal vents has been evaluated by using tested model for Selective (vibrational) Energy Transfer (SET).
The manuscript is clearly written and the data obtained are well presented and interpreted.
Suggested minor corrections:
1. Implement the quality of the submitted figures;
2. make English clearer
Author Response
Dear Referee 2,
Thank you for your positive attitude to our manuscript.
I understand that you raise two problems:
1. The quality of the figures that I use.
I am working to improve some of them; I should appreciate if you can inform the MDPI office which of the figures they regard the most urgent ones.
2. I am more than willing to have the language improved.
I herewith announce my partition in the MDPI language correction system. I hope to get an invoice to my home address to cover the costs.
Reviewer 3 Report
This study shows the effect of COS on the peptide synthesis. The reviewer has two questions about this manuscript and study, as shown below:
The calculation methods should be described in detail. What and how did the authors calculate and what was derived from such calculations? The method was not clearly shown in this paper.
Did the authors perform the calculation of HOMO and LUMO of the molecules using a quantum chemical method such as a DFT method? The reviewer considers that HOMO and LUMO calculations are very important in this kind of study.
Author Response
Dear Referee 3.
You ask two questions, and I shall try to answer them accordingly.
1. You ask how and what we calculated.
As far as I can understand we did not “calculate” anything.
What we have done was to describe the SET system which states that catalysis occurs as a transfer of energy from a certain vibration of the catalyst to a similar vibration in the reacting system. This latter vibration is supposed to be of such a kind that the “reacting molecule” is structurally disturbed so that it is forced to react.
Additionally, it is understood that this transfer of energy requires that the two vibrations described above are in a state of resonance ( e.g., so that the vibrational frequencies are alike, or - as might happen – one of the frequencies is twice, or thrice, etc, of the other).
This approach does not require any calculation, but only a strict comparison of observed (IR) spectroscopic data.
As is stressed in the manuscript, the concept of SET has been developed during the last 4 – 5 decades; and tested by physical measurements, where such could be done ( see section 2 , below).
But for the problems raised here, there are no targets for calculation. Hence, I have not made any alterations in the manuscript.
2. You ask about HOMO and LUMO calculations
I am not quite sure why this question is raised, but it leads the discussion to its root; At the end of the 19thcentury, the high point of classical physics, people were interested in energy transfer from one (mechanically) vibrating system to another one.
Under certain conditions, the rate of such a transfer was calculated, and a certain formalism was developed, Eqn [1] in the present paper. This formula has later on been extended to fit the formalism of quantum theory. In previously published papers [ e.g. 13-15, 17 of the manuscript ] applied to heterogeneous catalysis, we have integrated the rate expression over all values of the ‘quality factor’ ( see ref [13]). This integration was motivated by the many small variations that occur in a heterogeneous system . The resulting rate was put as equal to the rate of the chemical reaction.
An interesting consequence of this action is that it has been possible to determine the so called “isokinetic temperature”, Tiso, i.e., the temperature where the Arrhenius lines of several similar but not quite identical systems intersect in one precise point. The existence of this “isokinetic temperature” has been doubted, in some cases even hackled, but SET gives full “support”.
As an example I can quote an investigation;
M A Keane and R Larsson, “Isokinetic behavior in gas phase catalytic hydrodechlorination of chlorobenzene over supported nickel”, J. Mol. Catal.,2007, 268, 87-94.
As read from the Arrhenius lines we got Tiso= 669 ±2 K, whereas calculation from the SET formula gave Tiso= 669.2 K .
So, calculations and SET catalysis fit together, O K, but in the present case no calculations are needed; comparison of OBSERVED frequencies is all that is required.
Consequently, I have not performed any changes in the manuscript.
Round 2
Reviewer 3 Report
The reviewer understood the author's claim and has no further comments.